# Detection of *Coxiella burnetii* Using Silicon Microring Resonator in Patient Blood Plasma

**DOI:** 10.3390/mi10070427

**Published:** 2019-06-27

**Authors:** Bonhan Koo, Choong Eun Jin, Moonsuk Bae, Yoon Ok Jang, Ji Yeun Kim, Sung-Han Kim, Yong Shin

**Affiliations:** 1Department of Convergence Medicine, Asan Medical Institute of Convergence Science and Technology, Asan Medical Center, University of Ulsan College of Medicine, Seoul 05505, Korea; 2Department of Infectious Diseases, Asan Medical Center, University of Ulsan College of Medicine, Seoul 05505, Korea

**Keywords:** blood plasma, infectious diseases, biomolecular sensors, silicon microring resonator, acute Q fever

## Abstract

Blood plasma from patients is a powerful resource for diagnosing infectious disease due to it having many genetic materials as well as being relatively easy to obtain. Thus, various biosensors have been investigated for diagnosing diseases in blood plasma. However, there are no optimized and validated sensors for clinical use due to the low sensitivity, complexity, and difficulties of removing the inhibitors from plasma samples. In this study, we described a silicon microring resonator sensor used to detect *Coxiella burnetii* from the blood plasma of Q-fever patients in a label-free, real-time manner. Q-fever is an infectious disease caused by *Coxiella burnetii* via direct contact or inhalation aerosols. We validated this biosensor in the blood plasma of 35 clinical samples (including 16 Q fever samples infected with *Coxiella burnetii* and 19 samples infected with other febrile diseases. The biosensors are capable of rapid (10 min), highly sensitive (87.5%), and specific (89.5%) detection in plasma samples compared to the use of the conventional method.

## 1. Introduction

Non- or less invasive methods for collecting human fluidic specimens, such as blood, urine, saliva, and swab, are cost-effective, fast, and can be used repeatedly for a variety of infectious disease diagnoses reflecting the patient’s current status [1,2,3,4]. In particular, because blood is a rich source of genetic material, disease marker, and infectious agent, diagnosis using biomolecules of the blood is a powerful approach for the detection of infectious diseases [5,6]. Commonly, the four heme groups of hemoglobin present in red blood cells (RBCs) inhibit genetic analysis and diagnosis through their iron, so blood plasma is used to separate red blood cells from whole blood through centrifugation [7,8]. Even in the efforts to eliminate the inhibitors, including hemoglobin, IgG fraction, leukocyte DNA, and anticoagulants such as ethylenediaminetetraacetic acid (EDTA), heparin, and, sodium citrate, which are present in the plasma, these inhibitors act as an impediment to the diagnosis of infectious diseases, leading to an increased false-negative rate and low detection sensitivity [9,10,11,12].Therefore, diagnostic techniques are needed that can overcome low sensitivity and non-specific reactions as they are unaffected by various inhibitors present in the blood plasma.

Conventional methods for diagnosing Q fever typically use antibodies or nucleic acids detection methods. Antibody-based methods include indirect immunofluorescence [13] and enzyme-linked immunosorbent assay (ELISA) [14]. These methods have several limitations, such as a low sensitivity and specificity [13,14]. Nucleic acids-based methods use an end-point polymerase chain reaction (PCR) and real-time PCR system for the direct detection of *Coxiella burnetii* [15,16]. Although the PCR-based methods have a relatively high sensitivity and specificity compared to the antibody-based method, they require large equipment, such as a temperature control device and expensive dyes such as SYBR green and ethidium bromide (EtBr) dyes [15,16]. Recently many techniques for pathogen detection, using mechanical, electrical, electrochemical, and optical sensors, for easy to use, rapid, portable, multiplexed, and cost-effective pathogenic detection, have been developed [17]. They can feature high-throughput testing, increasing the efficiency of infectious disease diagnostics with a high sensitivity and specificity in laboratory testing level. One of these is based on mechanical sensors, the Quartz Crystal Microbalance (QCM) sensor, a label-free piezoelectric biosensor that measures the change in the resonance frequency caused by the increase of mass by attaching biomolecules to the sensor surface. The QCM sensor was able to detect very few bacterial cells and, in some cases, could detect down to 10 CFU/mL [18]. Another is based on electrochemical sensors, the amperometric biosensor, which is based on the direct measurement of the current produced by the oxidation or reduction of species by the interaction of biomolecules with biological receptors. Amperometric biosensors had a detection limit of 1 CFU/mL using a competitive magnetic immunoassay [19]. However, despite the advantages of these biosensors, there is no established method for detecting pathogens in blood plasma specimens.

In this work, we present a highly sensitive silicon microring resonator (SMR) bio-optical sensor based on isothermal nucleic acid amplification for the label-free detection of infectious agents using blood plasma specimens. Their operation is based on the change of the refractive index to the measurable spectral shift of the optical transmission, and they enable a real-time, label-free detection by monitoring changes in resonant wavelengths generated by biomolecules such as pathogens, proteins, and nucleic acids coupled with sensor ligands present on the sensor surface [20,21,22,23,24,25]. Photothermal spectroscopy, which indirectly measures the optical absorption of a material, allows measurements that are sensitive to changes in external conditions due to absorption only, unlike conventional methods of measuring the scattering and return loss [26]. SMR chips are fabricated using CMOS technology, which is widely used for bio-sensing applications due to the high quality and low cost when mass produced. SMR sensor technology, using extracted DNA from the blood plasma of infectious disease patients, shows that it is, however, possible to diagnose patients who are difficult to clinically diagnose quickly and in a real-time manner. Acute Q fever may progress to a persistent, intensive infection such as endocarditis if not initially treated, but it is difficult to diagnose because there are no distinct features that distinguish it from other febrile diseases [27,28]. In this study, we are developing a sensor based on SMR to detect the extracted DNA from 35 clinical samples (including 16 Q acute Q fever samples infected with *Coxiella burnetii* and 19 samples infected with other febrile diseases). Furthermore, we described several novelties regarding the SMR sensor for diagnosing Q fever compared to the previous study. In our previous proof-of-concept study, the SMR sensor was more sensitively developed for the detection of *C. burnetii* than conventional methods for Q fever diagnosis using frozen formaldehyde-fixed paraffin-embedded tissue and frozen blood plasma specimens from the Q fever patients [29,30]. On the other hand, in this study, we first optimized the sensor for a rapid and accurate diagnosis of Q fever in prospectively collected fresh blood plasma specimens (Figure 1). Second, we validated that the sensor can distinguish Q fever from other febrile diseases, which are showing similar symptoms with Q fever patients. Third, the detection time of the SMR sensor for diagnosing Q fever (10 min) was 20 min faster than that of the previous study (30 min). Fourth, we validated the clinical utility of the sensor in 35 patient samples. These results present that it can be applied to the diagnosis of diseases using clinical blood plasma in emergency patients with rapidity and specificity (Table 1).

## 2. Materials and Methods

### 2.1. Sensor Chip Fabrication and Surface Functionalization

The sensor chip uses SMR fabricated and described using previous protocols [21,22,23]. Briefly, the sensor array consists of four microrings, each with a spacing of 750 μm, and has one common input waveguide (through), and each microring has a dedicated output waveguide (drop). The sensor chip structures, such as microring structures, waveguides, and gratings, were patterned on a commercially available 200 mm Silicon-On-Insulator (SOI) wafer with a 220 nm thick top silicon layer, and a 2 μm thick buried oxide layer by 210 nm deep ultraviolet (UV) lithography. Light from an external near-IR tunable laser (1530–1570 nm) passes through an in-line polarization controller and is coupled to a vertical gray coupler located at one end of the linear waveguide. The insertion loss (IL) spectrum was measured using an EXFO IQS-2600B DWDM passive component test system (Figure 2). Next, the DNA primer immobilization process for the functionalization of the sensor, described in the previous protocol, was used [22,23]. Briefly, the functionalization of the sensor chips consists of four steps, the first step being the hyrdoxyl group modification on the chip surface using the oxygen plasma treatment. The second step is silanization, using 3–aminopropyltriethoxysiane (APTES, Sigma-Aldrich, St. Louis, MO, USA). The oxygen plasma treated sensor was immersed in a solution of 2% APTES in a mixture of ethanol–H2O (95:5, v/v) for two hours and was then thoroughly rinsed with ethanol and deionized (DI) water. The third step is modified with glutaraldehyde (GAD, Sigma-Aldrich) as an amine-amine crosslinking agent between amine functionalized modified sensors and a NH2 terminated DNA primer, in order to form an amide bond by covalent bonding on the surface of the sensor. The sensors were cured by heating them to 120 °C for 15 min. The sensors were then incubated with 2.5% GAD in DI water containing 5 mM sodium cyanoborohydride (Sigma-Aldrich) for 1 h, then rinsed with DI water and dried under a nitrogen (N2) stream. The last step is the immobilization of a NH2 terminated DNA probe. Pretreated sensors were prepared by incubation overnight (16 h) in 5 mM solutions of a DNA probe containing 5 mM sodium cyanoborohydride at room temperature. After incubation, unbound target specific primers were removed by washing them with 50 mM MES buffer, and the sensor was dried using an N2 steam. An acrylic well was then pasted onto the chip in order to enclose the microring sensor area. At this time, the chips were considered ready for the optical measurement. The functionalized sensor chips were sealed and stored at room temperature for later use. Basically, the wavelength without any molecules on the SMR sensor was not shifted at the initial wavelength value. However, when the desired molecule was attached to the sensor, the wavelength was shifted about 100–1000 pm depending on the target concentrations. Using this function, the SMR sensor could quantitatively detect the target molecules.

### 2.2. Blood Plasma Specimens

The 35 blood plasma samples from 16 acute Q fever patients infected with *Coxiella burnetii* (*C. burnetii*) and 19 other febrile diseases were obtained using protocols approved by the Institutional Review Board of Asan Medical Center (2018–9023), Republic of Korea [28]. We also obtained the institutional approval and written consent of the patient. To detect *C. burnetii*, DNA was extracted, using a QIAamp DNA Mini kit (Qiagen, Hilden, Germany) according to the manufacturer’s instructions, from the blood plasma of patients with suspected acute Q fever. We used blood plasma samples at a starting volume of 200 μL each and eluted approximately 100 μL using elution buffer. The extracted DNA was then aliquoted and stored at −20 °C until use. 

### 2.3. Amplification and Detection Using SMR Sensor

We prepared a recombinase polymerase amplification (RPA) solution capable of nucleic acid amplification under isothermal (37–42 °C) conditions in order to amplify and detect *C. burnetii* DNA. The forward and reverse primers for the SMR sensor were designed and synthesized at approximately 35 bp. The primers and RPA solutions used in the SMR sensor were prepared using the protocol described in a previous study (Table 2) [22]. To optimize the sensor surface nucleic acid amplification using the RPA solution, 29.5 μL of rehydration buffer, 15 μL of RNase inhibitor, DI water, 2 μM dethiothreitol (DTT), and 2.5 μL of 10 mM reverse primer were mixed, and the solution was added to one dried enzyme pellet. Then, 2.5 μL of magnesium acetate solution was dispensed into the cap of the tube. The unidirectional shake mode mixing protocol helps to homogeneously distribute the molecules required for the reactions present in the buffer. For use in the sensor reaction, only 10 μL of the 50 μL mixture solution is sampled and mixed with 5 μL of extracted DNA from the patient samples. 15 μL of the solution is filled into acrylic wells enclosing the microrings on the sensor, and the mineral oil is dispensed in order to prevent evaporation during amplification (Figure 2). To maintain the isothermal condition, a thermoelectric cooler (TEC) connected to a proportional integral differential controller (Alpha Omega Instruments, Lincoln, RI, USA) was used, and the resonance spectrum of the device was used immediately as a reference for obtaining the baseline. SMR allows the target DNA to selectively bind and amplify to the immobilized primer in the evanescent field of the resonator waveguide and then to increase the proportion of each wavelength. The wavelength shift was collected every 5 min for up to 20 min to monitor the amplification of the target DNA in a label-free and real-time manner. As an isothermal assay used for nucleic acid amplification, RPA technology does not require the initial denaturation step of target nucleic acid and was optimized at a relatively low temperature of 37–43 °C. In addition, it is also robust at off temperatures and low temperature setups, and generally operates at the typically ambient temperature of 25 °C. If the reaction temperature changes slightly or drops to the room temperature level, the SMR sensor assay takes a longer time for detection than the optimal temperature level, but it can detect target nucleic acid. In the SMR sensor, there is a sensing area consisting of three microrings for targets and one reference microring. The SMR sensor area is surrounded by an acrylic well and is filled with the reaction mixture containing RPA and target DNA for amplification on the sensing microring surface. Since the sensor area for the target detection is filled with the reaction mixture, the influence of humidity on the detection sensitivity is very small. SMR chips, after the primers immobilized on the surface of the sensor area, can be used until 2 days later, but we prefer to use the sensor on the same day for removing any influence on the detection sensitivity by unknown inhibitors. Then, the SMR sensor can be reused after the Piranha cleaning (70:30 H_2_SO_4_:H_2_O_2_) and repeated surface modification. 

## 3. Results and Discussion

### 3.1. Principle of SMR Biosensor

Figure 1 shows a schematic of the detection principle using the silicon microring resonator (SMR) sensor for DNA extracted from the blood plasma of the acute Q fever patients. The extracted DNA is amplified by the isothermal–based, recombinase polymerase amplification (RPA) reagent. The DNA primer immobilized in the SMR sensor is complementary to the target sequence, and was designed for RPA isothermal amplification. The amine (NH2) terminated, forward DNA primer was grafted on the sensor surface, and the reverse DNA primer was mixed with the extracted DNA and the RPA reagent in the acrylic well. The immobilized primer was hybridized with the target DNA on the SMR surface, and the target was amplified by the RPA reagent at 38 °C. The resonant wavelength is shifted over time by the amplified target product and measured in a label-free, real-time manner. Figure 3 shows the representative resonant wavelength shift for the detection of *C. burnetii* up to 20 min between the target from acute Q fever and other febrile diseased patients. The resonant wavelength shifts in 20 min using the SMR sensor were 429.41 pm ± 16.27 in the acute Q fever plasma sample (positive) and 170.33 pm ± 13.30 in other febrile disease samples (negative). The resonant wavelength shift in the presence of *C. burnetii* DNA can be higher than the detection criterion, which was established by the resonant wavelength obtained from the negative samples. As a result, the targets were detected in plasma samples by measuring the resonant wavelength shift within 10 min.

### 3.2. Sensitivity of SMR Sensor in Clinical Plasma Specimens

To verify the clinical sensitivity of the SMR biosensor, the clinical plasma samples obtained from 16 confirmed Q fever patients, and 19 other febrile diseases were used (Figure 4). Since Q fever does not differ significantly from other febrile diseases, the detection of acute Q fever through a rapid and accurate diagnosis is important [29]. We used the complementary DNA primer of the IS1111a gene of *C. burnetii* (NCBI Nr. M8806) in previous study [30] and measured the resonant wavelength shift assay every 5 min up to 20 min. The SMR biosensor detected 14 of 16 patients infected with 16 confirmed Q fever samples, and showed a high sensitivity of over 87.5% (14/16) in 10 min (Table 3). Notably, the SMR sensor is capable of Q fever rapid detection within 10 min, except the extraction step of DNA from the plasma samples. Nevertheless, the sensor is faster than the conventional end-point PCR method for pathogen detection (>2 h) [28]. Therefore, we have shown that this SMR biosensor can be useful for pathogen detection in blood plasma samples.

### 3.3. Specificity of SMR Sensor in Clinical Plasma Specimens

To identify the clinical specificity of the SMR biosensor, we used a clinical plasma sample from each of 19 patients with other febrile diseases, as well as the complementary DNA primer of the IS1111a gene of *C. burnetii* (Figure 4). We also measured the resonant wavelength shift every 5 min for as long as 20 min using clinical specimens of other febrile diseases. The SMR biosensor confirmed that *C. burnetii* was not detected in 17 of 19 other febrile samples and showed a high specificity of 89.5% (17/19) at 10 min (Table 3). Therefore, the SMR biosensors have a higher specificity when using plasma samples from other febrile diseases. Our SMR biosensor showed a superior ability to detect pathogens in actual plasma samples and with a clinical utility.

## 4. Conclusions

In this study, we proposed a rapid and accurate diagnosis technique for infectious disease in plasma samples using the SMR sensor. Our system showed the real-time and label-free detection of DNA from *Coxiella burnetii* using plasma samples from Q fever patients In addition, a small amount of reaction volume (a total of 15 μL volume contained RPA reaction buffer and 5 μL of extracted DNA) was compared to that of the conventional PCR assay (a total of 25 μL volume contained PCR reaction buffer and 5 μL of extracted DNA), no thermoregulator was required and it was possible to determine positive or negative within 10 min. Our system has several advantages in plasma-based diagnostic. First of all, it is suitable for diagnosis using clinical samples, such as plasma, obtained from the patients. Previously reported plasma-based infectious disease diagnoses using the conventional PCR method showed a low sensitivity and specificity due to the various inhibitors contained in plasma, thus requiring an additional purification step after the DNA extraction because it was difficult to accurately diagnose the infected patients [31]. This study confirms the clinical utility of 35 clinical plasma samples without additional purification steps after the DNA extraction using the SMR sensor. The sensitivity of our SMR sensor (87.5%) with blood plasma samples was higher than the PCR assay (81.3%), and it was confirmed that the SMR sensor showed a similar specificity (89.5%) compared to the PCR assay. The immobilization of the DNA probe and amplification of the DNA product on the surface can reduce the non-specific reactions caused by primers present in the liquid phase, thereby increasing the detection specificity [32]. These results show that the SMR sensor can be used as a substitute for the existing plasma-based diagnostic method because it can be detected more quickly (< 10 min) than the PCR assay (> 2 h) and also has a high sensitivity. Despite the enhancement of the current SMR sensor (Table 1), the validation of the sensor must be verified by using plasma samples from patients with various infectious diseases and various symptoms. Therefore, further development is required in order to process a large number of plasma samples through the scaling-up of the SMR sensor or in order to be able to process a variety of biomarkers at the same time.

## Figures and Tables

**Figure 1 micromachines-10-00427-f001:**
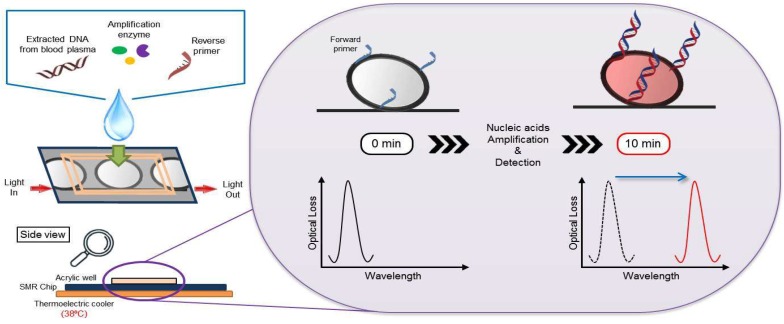
Schematic representation of the principle of isothermal nucleic acid amplification and detection using a silicon microring resonator sensor in clinical blood plasma specimens. In the isothermal condition (38 °C), the DNA is hybridized with immobilized DNA primer and the target sequence is amplified by the RPA reagent. The resonant wavelength is shifted over time by nucleic acid amplification on the sensor microring and measured in a real-time manner.

**Figure 2 micromachines-10-00427-f002:**
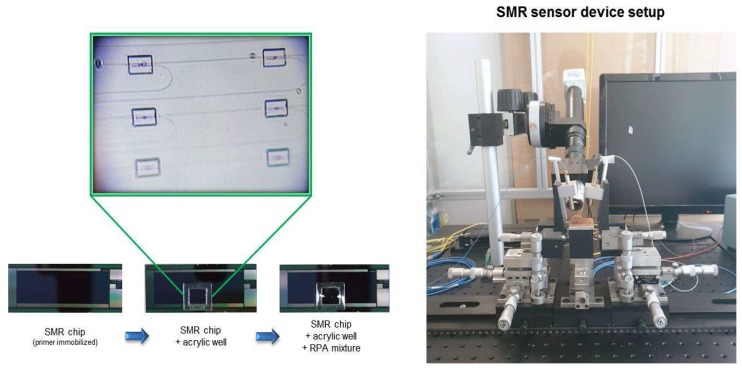
Images of the SMR chip and SMR sensor device setup.

**Figure 3 micromachines-10-00427-f003:**
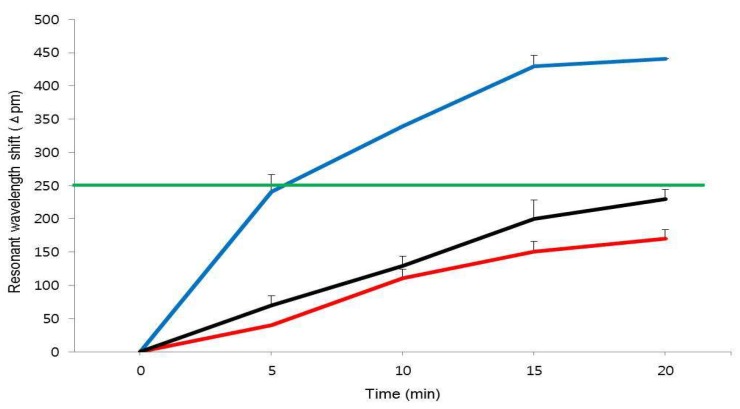
The identification resonant wavelength shift for the *C. burnetii* detection using a silicon microring resonator sensor. The blue line shows an acute Q fever specimen (*C. burnetii* DNA) as a positive, and the red and dark lines indicate the other febrile disease specimen and healthy control (no infection) as negatives, respectively. The resonant wavelength shift for the detection of *C. burnetii* was shown within 20 min. The green line indicates a criterion for positive and negative determination in 20 min. The error bars indicate the standard deviation of the mean, based on at least three independent experiments.

**Figure 4 micromachines-10-00427-f004:**
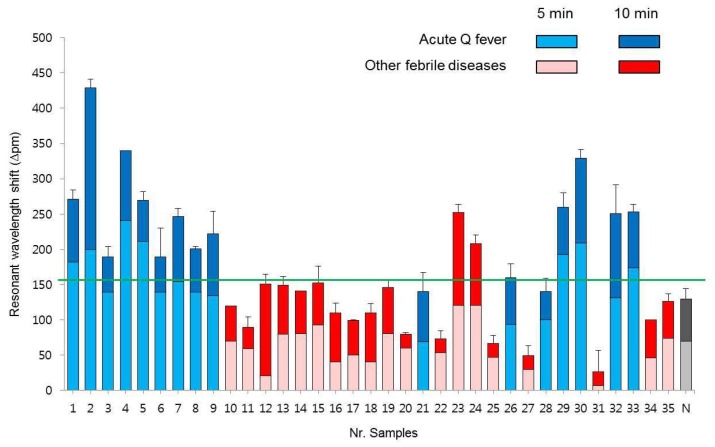
Clinical utility of the silicon microring resonator biosensor with 35 patient blood plasma specimens for the *C. burnetii* detection. The bar graph shows the resonant wavelength shift in 5 and 10 min. The blue bars indicate the result of 5 min (light blue) and 10 min (blue) of 16 samples of acute Q fever patients, and the red bars indicate the result of 5 min (light red) and 10 min (red) of 19 samples of other febrile diseases patients, while the gray bar indicates the result of 5 min (light gray) and 10 min (gray) of the healthy control. The green line indicates a criterion for positive and negative determination in 10 min. The error bars indicate the standard deviation of the mean, based on at least three independent experiments.

**Table 1 micromachines-10-00427-t001:** Comparison of the current and previous system of the silicon microring resonator (SMR) sensor.

Content	Current	Previous [30]
Target diseases	Q fever	Q fever Tick-borne diseases Respiratory virus infection diseases
Clinical specimen	Fresh blood plasma	Frozen tissue Frozen blood plasma Nasopharyngeal
Amplification condition	38 °C (for DNA)	38 °C (for DNA) 43 °C (for RNA)
Limit of detection (copies/reaction)	10^1^–10^2^	Not Tested
Reaction time	20 min	30 min
Detection time	10 min	20–30min

**Table 2 micromachines-10-00427-t002:** Primer sequences of *C. burnetii.*

*C. burnetii*	Primer	Sequence
IS1111a *	End-point PCR—Forward	5’-GAGCGAACCATTGGTATCG-3’
	End-point PCR—Reverse	5’-CTTTAACAGCGCTTGAACGT-3’
	SMR sensor—Forward	5’-NH2-(CH2)12-GAGCGAACCATT GGTATCGGACGTTTATGGGGATG-3’
	SMR sensor—Reverse	5’-GTATCTTTAACAGCGCTTGAACGTCTTGTTG-3’

* IS1111a; *C. burnetii* transposase (IS1111a) gene, complete coding sequence (NCBI Nr. M8806).

**Table 3 micromachines-10-00427-t003:** The clinical sensitivity and specificity of the SMR biosensor for Q fever diagnosis using 35 clinical specimens at 10 min.

SMR Sensor				End-Point PCR [28]			
Content	Clinical Positive	Clinical Negative	Total	Content	Clinical Positive	Clinical Negative	Total
Test Positive	14	2	16	Test Positive	13	2	15
Test Negative	2	17	19	Test Negative	3	17	20
-	16	19	-	-	16	19	-
-	Sensitivity 87.5%	Specificity 89.5%	-	-	Sensitivity 81.3%	Specificity 89.5%	-

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
