# Peer review of "Detection of Coxiella burnetii Using Silicon Microring Resonator in Patient Blood Plasma"

_micromachines, 2019, doi:10.3390/mi10070427_

Round 1
Reviewer 1 Report
General comments:
(N1) You did not answer on the manuscript to my previous comment 1: (1) A lot of information is presented in past references of the authors, such as 16-18 and 24 and 26, and in the manuscript it cannot be understood the improvements regarding to these previous publications. This should be clearly explained in the manuscript.
As the previously version of the manuscript, you do not refer in the manuscript what are the differences between this one and the previously published 22-24. You told to the reviewer in the letter, but the readers will not read the letter, only the manuscript. This should be clearly state in the manuscript. Like that it can not be published.
(N2) in the previous revision it is stated: (3) The introduction is not well performed. It should refer more the SMR sensors, if there is more related and for what detection values they exist. Their performance. What advantages and disadvantages of its use in this application.
You answer this adding information on table3. This information is relevant but it was not what I’ve asked. I’ve asked for SMR sensors of literature for technologic and performance comparison with yours.
(N3) in the previous revision it is stated: (4) The used volumes for the detection are similar to the conventional methods? Or it uses less?
You answer to the review but I did not see this information on the manuscript.
(N4) in the previous revision it is stated: (6) A photo of the fabricated biosensor should be presented, once it is claimed that it detects.
This photo must have more zoom to effectively see the sensor area.
Correct the word sensor in the figure 2 caption.
(N5) in the previous revision it is stated: (7) What about if the temperature and humidity conditions when the test is done (the test of figure3) changes? The results would be the same? What will vary? For how long it is stable the SMR biosensor: lifetime before use?
Once again, you answer to the review, but you did not introduce the information in the manuscript.
Specific comments:
Page 1, line 33-35: this sentence has not a conclusion and it starts as Despite. You must rephrase this sentence.
Page 2, line 45: you should remove the word "but", and you should remove the word "requires"
Page 2, line 47: “… using sensors …” you can not tell this as it is. The conventional methods also use sensors even using a big and expensive equipment but, for reading the values, it is always need a sensor. You must re-phase this beginning.
For example: Recently, many techniques for pathogens detection using mechanical, electrical, electrochemical, and optical sensors, for easy to use, rapid, portable, multiplexed, and cost-effective pathogenic detection, have been developed [17]. They can feature high-throughput testing, increasing the efficiency of infectious disease diagnostics with high sensitivity and specificity in laboratory testing level. One of those, based ion mechanical sensors, the Quartz .....
Page 2, line 49-50: re-phase this sentence, after the comma the sentence has no connection. It is missing some word. And also the beginning with the word biosensors it fall from sky.
Page 2, line 54: add commas: "... and, in some cases, ..." ; Change the sentence to: "Another, based on electrochemical ...."
Page 2, line 55: “… sensors,” sensors, specifically, amperometric ... and they are sensors or biosensors????
Page2, line 57: in the pevious sentence you refer direct measurement of current and now you state Indirect amperometric .... despite not being the same thing, it should be clearly. Some readers are not completely expert on electrochemical measurements.
Page 2, line 58: “sensors” are they sensors or biosensors? It is different.
Page 2, line 64: Change the beginning of this sentence to: "Their operation is based on the change of the refractive index to the ....."
Page 2, line 72-73: “In addition, various targets can be detected by integrating sensors 72 measuring various biomarkers. [20].” you should remove this sentence, here it makes no sense.
Page 2, line 72: Change to: low cost when mass production
Page 2, line 74-79: this paragraph should be before talk about your sensor. Because this is the novelty of your work, and because of that, you are developing a sensor that .....
Table 1: This table should be placed near it is referred, which is 2 pages after.
Author Response
Responses to reviewer’s comments
We thank the reviewers for their thoughtful critique of our manuscript. We have carefully taken their comments into consideration when preparing our revised manuscript, which has improved significantly the quality of the manuscript. The following text shows our point-by-point responses.
Reviewer #1
(N1) You did not answer on the manuscript to my previous comment 1: (1) A lot of information is presented in past references of the authors, such as 16-18 and 24 and 26, and in the manuscript it cannot be understood the improvements regarding to these previous publications. This should be clearly explained in the manuscript.
As the previously version of the manuscript, you do not refer in the manuscript what are the differences between this one and the previously published 22-24. You told to the reviewer in the letter, but the readers will not read the letter, only the manuscript. This should be clearly state in the manuscript. Like that it can not be published.
Ø We thank the reviewer for the comment. We have added the information in the conclusions of the revised manuscript..
(N2) in the previous revision it is stated: (3) The introduction is not well performed. It should refer more the SMR sensors, if there is more related and for what detection values they exist. Their performance. What advantages and disadvantages of its use in this application.
You answer this adding information on table3. This information is relevant but it was not what I’ve asked. I’ve asked for SMR sensors of literature for technologic and performance comparison with yours.
Ø We thank the reviewer for the comment. We have added the information on Table 3 in the revised manuscript as followed.
Current | Previous[30] | |
Target diseases | Q fever | Q fever Tick-borne diseases Respiratory virus infection diseases |
Clinical specimen | Fresh blood plasma | Frozen tissue Frozen blood plasma Nasopharyngeal |
Amplification condition | 38 °C (for DNA) | 38 °C (for DNA) 43 °C (for RNA) |
Limit of detection (copies/reaction) | - | Not Tested |
Reaction time | 20 min | 30 min |
Detection time | 10 min | 20-30min |
(N3) in the previous revision it is stated: (4) The used volumes for the detection are similar to the conventional methods? Or it uses less?
You answer to the review but I did not see this information on the manuscript.
Ø We thank the reviewer for the comment. We have added the information in the conclusions of the revised manuscript.
In Conclusions, “In addition, a small amount of reaction volume (total of 15 μL volume contained RPA reaction buffer and 5 μL of extracted DNA) is compared to that of the conventional PCR assay (total of 25 μL volume contained PCR reaction buffer and 5 μL of extracted DNA)……..”
(N4) in the previous revision it is stated: (6) A photo of the fabricated biosensor should be presented, once it is claimed that it detects.
This photo must have more zoom to effectively see the sensor area.
Ø We thank the reviewer for the comment. According to the reviewer’s comment, we have corrected it at the Figure 2 in the revised manuscript.
Correct the word sensor in the figure 2 caption.
Ø We thank the reviewer for the comment. According to the reviewer’s comment, we have corrected it in the revised manuscript.
(N5) in the previous revision it is stated: (7) What about if the temperature and humidity conditions when the test is done (the test of figure3) changes? The results would be the same? What will vary? For how long it is stable the SMR biosensor: lifetime before use?
Once again, you answer to the review, but you did not introduce the information in the manuscript.
Ø We thank the reviewer for the comment. We have added the information on 2.3 section in the revised manuscript.
Specific comments:
Page 1, line 33-35: this sentence has not a conclusion and it starts as Despite. You must rephrase this sentence.
Ø We thank the reviewer for the comment. According to the reviewer’s comment, we have modified the sentence in the revised manuscript.
Page 2, line 45: you should remove the word "but", and you should remove the word "requires"
Ø We thank the reviewer for the comment. According to the reviewer’s comment, we have modified those in the revised manuscript.
Page 2, line 47: “… using sensors …” you can not tell this as it is. The conventional methods also use sensors even using a big and expensive equipment but, for reading the values, it is always need a sensor. You must re-phase this beginning.
For example: Recently, many techniques for pathogens detection using mechanical, electrical, electrochemical, and optical sensors, for easy to use, rapid, portable, multiplexed, and cost-effective pathogenic detection, have been developed [17]. They can feature high-throughput testing, increasing the efficiency of infectious disease diagnostics with high sensitivity and specificity in laboratory testing level. One of those, based ion mechanical sensors, the Quartz .....
Ø We thank the reviewer for the comment. According to the reviewer’s comment, we have modified the sentences in the revised manuscript.
Page 2, line 49-50: re-phase this sentence, after the comma the sentence has no connection. It is missing some word. And also the beginning with the word biosensors it fall from sky.
Ø We thank the reviewer for the comment. We have deleted the sentence in the revised manuscript.
Page 2, line 54: add commas: "... and, in some cases, ..." ; Change the sentence to: "Another, based on electrochemical ...."
Ø We thank the reviewer for the comment. According to the reviewer’s comment, we have modified the sentence in the revised manuscript.
Page 2, line 55: “… sensors,” sensors, specifically, amperometric ... and they are sensors or biosensors????
Ø We thank the reviewer for the comment. According to the reviewer’s comment, we have modified the sentence in the revised manuscript.
Page2, line 57: in the pevious sentence you refer direct measurement of current and now you state Indirect amperometric .... despite not being the same thing, it should be clearly. Some readers are not completely expert on electrochemical measurements.
Ø We thank the reviewer for the comment. According to the reviewer’s comment, we have corrected the sentence in the revised manuscript.
Page 2, line 58: “sensors” are they sensors or biosensors? It is different.
Ø We thank the reviewer for the comment. According to the reviewer’s comment, we have corrected the sentence in the revised manuscript.
Page 2, line 64: Change the beginning of this sentence to: "Their operation is based on the change of the refractive index to the ....."
Ø We thank the reviewer for the comment. According to the reviewer’s comment, we have modified it in the revised manuscript.
Page 2, line 72-73: “In addition, various targets can be detected by integrating sensors 72 measuring various biomarkers. [20].” you should remove this sentence, here it makes no sense.
Ø We thank the reviewer for the comment. According to the reviewer’s comment, we have deleted it in the revised manuscript.
Page 2, line 72: Change to: low cost when mass production
Ø We thank the reviewer for the comment. According to the reviewer’s comment, we have corrected it in the revised manuscript.
Page 2, line 74-79: this paragraph should be before talk about your sensor. Because this is the novelty of your work, and because of that, you are developing a sensor that .....
Ø We thank the reviewer for the comment. According to the reviewer’s comment, we have corrected it in the revised manuscript.
Table 1: This table should be placed near it is referred, which is 2 pages after.
Ø We thank the reviewer for the comment. According to the reviewer’s comment, we have moved it before Section 3 in the revised manuscript.
Reviewer 2 Report
Authors answered most of Reviewer's questions adequately. The manuscript can be considered for publication after minor revision.
Authors state on Page 4 that “The SMR sensor can be reused after the Piranha cleaning (70: 30 H2SO4 : H2O2).” Please add "and repeated surface functionalization."
Author Response
Responses to reviewer’s comments
We thank the reviewers for their thoughtful critique of our manuscript. We have carefully taken their comments into consideration when preparing our revised manuscript, which has improved significantly the quality of the manuscript. The following text shows our point-by-point responses
Reviewer #2
Authors answered most of Reviewer's questions adequately. The manuscript can be considered for publication after minor revision.
Authors state on Page 4 that “The SMR sensor can be reused after the Piranha cleaning (70: 30 H2SO4 : H2O2).” Please add "and repeated surface functionalization."
Ø We thank the reviewer for the comment. According to the reviewer’s comment, we have added it in the revised manuscript.
Reviewer 3 Report
The paper has been revised and is now improved. I think it can be accepted if the authors make the following changes:
- In the abstract (lines 19-20) and in the introduction (lines 73-74) it is stated that the sensor has been validated with 35 samples of Q fever patients infected by C burnetii. However, in Section 2.2 (lines 124-125) and in section 3.2 (lines 181-182) as well as in Figure 4 caption it is stated that the complete set of 35 plasma samples is composed of 16 Q fever samples and 19 samples infected with other febrile diseases. I think the sentences in the abstract and introduction should be corrected.
- In Figure 3 caption (lines 177-178) there is the sentence “The green line indicates a criterion for positive and negative determination in 20 min”. I think it is “10 min”.
- Some typos must be corrected: line 43 “burntii” must be “burnetii”; line 85 “750 um” must be “750 µm”; line 144 “15 µl” must be “15 µL”.
Author Response
Responses to reviewer’s comments
We thank the reviewers for their thoughtful critique of our manuscript. We have carefully taken their comments into consideration when preparing our revised manuscript, which has improved significantly the quality of the manuscript. The following text shows our point-by-point responses
Reviewer #3
The paper has been revised and is now improved. I think it can be accepted if the authors make the following changes:
- In the abstract (lines 19-20) and in the introduction (lines 73-74) it is stated that the sensor has been validated with 35 samples of Q fever patients infected by C burnetii. However, in Section 2.2 (lines 124-125) and in section 3.2 (lines 181-182) as well as in Figure 4 caption it is stated that the complete set of 35 plasma samples is composed of 16 Q fever samples and 19 samples infected with other febrile diseases. I think the sentences in the abstract and introduction should be corrected.
Ø We thank the reviewer for the comment. According to the reviewer’s comment, we have added it in the abstract and introduction of the revised manuscript.
- In Figure 3 caption (lines 177-178) there is the sentence “The green line indicates a criterion for positive and negative determination in 20 min”. I think it is “10 min”.
Ø We thank the reviewer for the comment. It is correct that the criterion was selected for positive and negative determination in 20 min.
- Some typos must be corrected: line 43 “burntii” must be “burnetii”; line 85 “750 um” must be “750 µm”; line 144 “15 µl” must be “15 µL”.
Ø We thank the reviewer for the comment. According to the reviewer’s comment, we have corrected those in the revised manuscript.
Round 2
Reviewer 1 Report
(N1) The information you added in the conclusion, it should be in the introduction for the reader see at the beginning of the manuscript, the differences between this one and your previously published 22-24. You must change to the introduction.
(N2) you introduces the information on conclusion. This should be introduced in the introduction for the reader see at the beginning the what is being done in the state of the art about SMR sensors. Again, you must change to the introduction.
Page 2, line 47: You should include a reference where it is seen that there exist: easy to use, rapid and portable lab-on-a-chips for other analysis, to see that is viable your subject, and you will perform for your application. A good reference is: Biosensors and Bioelectronics (2017) 90, pp,. 308-313.
(N4) Correct the word sensor in the figure 2 caption.
Ø We thank the reviewer for the comment. According to the reviewer’s comment, we have corrected it in the revised manuscript.
It is not yet corrected, it is “Figure 2. Images of SMR chip and SMR senor device setup.” And it is not senor but sensor.
Author Response
Responses to reviewer’s comments
We thank the reviewers for their thoughtful critique of our manuscript. We have carefully taken their comments into consideration when preparing our revised manuscript, which has improved significantly the quality of the manuscript. The following text shows our point-by-point responses.
Reviewer #1
(N1) The information you added in the conclusion, it should be in the introduction for the reader see at the beginning of the manuscript, the differences between this one and your previously published 22-24. You must change to the introduction.
Ø We thank the reviewer for the comment. According to the reviewer’s comment, we have moved the information to the introduction part of the revised manuscript..
(N2) you introduces the information on conclusion. This should be introduced in the introduction for the reader see at the beginning the what is being done in the state of the art about SMR sensors. Again, you must change to the introduction.
Ø We thank the reviewer for the comment. According to the reviewer’s comment, we have moved the information to the introduction part and Table 1 of the revised manuscript.
Page 2, line 47: You should include a reference where it is seen that there exist: easy to use, rapid and portable lab-on-a-chips for other analysis, to see that is viable your subject, and you will perform for your application. A good reference is: Biosensors and Bioelectronics (2017) 90. Pp. 308-313.
Ø We thank the reviewer for the comment. According to the reviewer’s comment, we have added it in the revised manuscript.
(N4) Correct the word sensor in the figure 2 caption.
Ø We thank the reviewer for the comment. According to the reviewer’s comment, we have corrected it in the revised manuscript.